# Learning Credal Sum-Product Networks[*]

**Amélie Levray**                                                        ALEVRAY@INF.ED.AC.UK
*University of Edinburgh, UK*

**Vaishak Belle**                                                        VAISHAK@ED.AC.UK
*University of Edinburgh, UK & Alan Turing Institute, UK*

## Abstract

Probabilistic representations, such as Bayesian and Markov networks, are fundamental to much of statistical machine learning. Thus, learning probabilistic representations directly from data is a deep challenge, the main computational bottleneck being inference that is intractable. Tractable learning is a powerful new paradigm that attempts to learn distributions that support efficient probabilistic querying. By leveraging local structure, representations such as sum-product networks (SPNs) can capture high tree-width models with many hidden layers, essentially a deep architecture, while still admitting a range of probabilistic queries to be computable in time polynomial in the network size. While the progress is impressive, numerous data sources are incomplete, and in the presence of missing data, structure learning methods nonetheless revert to single distributions without characterizing the loss in confidence. In recent work, credal sum-product networks, an imprecise extension of sum-product networks, were proposed to capture this robustness angle. In this work, we are interested in how such representations can be learnt and thus study how the computational machinery underlying tractable learning and inference can be generalized for imprecise probabilities.

## 1. Introduction

Probabilistic representations, such as Bayesian and Markov networks, are fundamental to much of statistical machine learning. Thus, learning probabilistic representations directly from data is a deep challenge. Unfortunately, exact inference in probabilistic graphical models is intractable [Valiant, 1979, Bacchus et al., 2009]. Naturally, then, owing to the intractability of inference, learning also becomes challenging, since learning typically uses inference as a sub-routine [Koller and Friedman, 2009]. Moreover, even if such a representation is learned, prediction will suffer because inference has to be approximated. Tractable learning is a powerful new paradigm that attempts to learn representations that support efficient probabilistic querying. Much of the initial work focused on low tree-width models [Bach and Jordan, 2002], but later, building on properties such as local structure [Chavira and Darwiche, 2008], data structures such as arithmetic circuits (ACs) emerged. These circuit learners can also represent high tree-width models and enable exact inference for a range of queries in time polynomial in the circuit size. Sum-product networks (SPNs) [Poon and Domingos, 2011] are instances of ACs with an elegant recursive structure – essentially, an SPN is a weighted sum of products of SPNs, and the base case is a leaf node denoting a tractable probability distribution (e.g., a univariate Bernoulli distribution). In so much as deep learning models can be understood as graphical models with multiple hidden variables, SPNs can be seen as a tractable deep architecture. Of course, learning the architecture of standard deep models is very challenging [Bengio, 2009], and in contrast, SPNs, by their very design, offer a reliable structure learning paradigm. While it is possible to specify SPNs by hand, weight learning is additionally required to obtain a probability distribution, but also the specification of SPNs has to obey conditions of completeness and decomposability, all of which makes

---

∗. This work was supported by the EPSRC grant *Towards Explainable and Robust Statistical AI: A Symbolic Approach.* Vaishak Belle was also partly supported by a Royal Society University Research Fellowship.

structure learning an obvious choice. Since SPNs were introduced, a number of structure learning frameworks have been developed for these and related data structures, e.g., [Gens and Domingos, 2013, Hsu et al., 2017, Liang et al., 2017]. Such structures are also turning out to be valuable in relational settings: on the one hand, they have served as compilation targets for relational representations [Fierens et al., 2011, Chavira et al., 2006], and on the other, they have also been extended with a relational syntax [Nath and Domingos, 2015, Van den Broeck et al., 2011].

The question that concerns us in this work is handling incomplete knowledge, including missing data. Classically, the issue of missing values or missing tuples is often ignored, and structure learning methods nonetheless revert to single distributions without characterizing the loss in confidence. Indeed, deletion and naive imputation schemes can be particularly problematic [Van Buuren, 2018]. Imprecise probability models augment classical probabilistic models with representations for incomplete and indeterminate knowledge [Augustin et al., 2014, Halpern, 2003]. Credal networks [Cozman, 2000], for example, extend Bayesian networks in enabling sets of conditional probability measures to be associated with variables.

The benefits of such models are many. As argued in [Mauá et al., 2017], learning single distributions in the presence of incomplete knowledge may led to predictions that are unreliable and overconfident. By allowing imprecise probability estimates, there is the added semantic value of knowing that the distribution of a random variable is not certain, which indicates to the user that her confidence in predictions involving the said random value should be appropriately adjusted. Furthermore, a quantified assessment of this confidence is possible. Conversely, if we have a pre-trained model but whose data sources are known to have had missing information, we may consider contamination strategies to capture the unreliability of our predictions [Mauá et al., 2017]. In the context of probabilistic databases [Suciu et al., 2011], the value of such representations have also been argued in [Ceylan et al., 2016] as an elegant alternative to enforcing the closed-world assumption (CWA) [Reiter, 1977]. Recall that in probabilistic databases, all tuples in the database are accorded a confidence probability (resulting from information extraction over natural language sources, for example), but all other facts have probability zero. The CWA is inappropriate because sources from which such representations are obtained are frequently updated, and it is clearly problematic to attach a prior probability of zero to some fact that we later want to consider plausible. In that regard, [Ceylan et al., 2016] consider extending probabilistic databases with credal notions to yield an *open-world* representation: all tuples not in the database take on probabilities from a default probability interval, corresponding to lower and upper probabilities, which can then be updated on observing such facts in newer sources.[1] Thus, we obtain a principled means of handling "unseen" observations. Analogously, when computing the probability of queries, the representation allows the user to recognise that there is incomplete knowledge, and so she can use interval probabilities for assessing the degree of confidence in a returned answer. Overall, the credal representation can help us model, query and learn with incomplete data in a principled manner. Since we expect most sources of data to be incomplete in this sense of discovering unseen facts from new and updated sources, the benefits are deeply significant.

In the context of tractable models, [Mauá et al., 2017] proposed a unification of the credal network semantics and SPNs, resulting in so-called credal SPNs, and study algorithms and complexity results for inference tasks. In this work, we are interested in how such representations can be learnt and thus study how the computational machinery underlying tractable learning and inference can be generalized for imprecise probabilities. We report on both formal and empirical results on that investigation, with very promising ob-

---

1. This perspective where the domain of constants is known in advance can be contrasted with a second notion of "open"-ness that does not require the constants to be known in advance [Russell, 2015, Belle, 2017, Grohe and Lindner, 2019]. In these latter formulations, the universe is often countably infinite.

servations.[2] We hope this work would be useful for the emerging interest in handling incomplete knowledge and missing data for large-scale machine learning.

## 2. Preliminaries

### 2.1 Sum-product Networks (SPNs)

SPNs are rooted acyclic graphs whose internal nodes are sums and products, and leaf nodes are tractable distributions, such as Bernoulli and Gaussian distributions [Poon and Domingos, 2011, Gens and Domingos, 2013]. More formally,

**Definition 1.** [Syntactic definition] An SPN over the set of variables $X_1, ..., X_n$ is a rooted directed acyclic graph whose leaves are the indicators $x_1, ..., x_n$ and $\overline{x}_1, ..., \overline{x}_n$ and whose internal nodes are sums and products nodes. Each edge $(i, j)$ from a sum node $i$ has a non-negative weight $\omega_{ij}$.

**Definition 2.** [Semantic definition] The value of a product node is the product of the values of its children. The value of a sum node is $\sum_{j \in Ch(i)} \omega_{ij} \upsilon_j$, where $Ch(i)$ are the children of $i$ and $\upsilon_j$ is the value of node $j$. The value of an SPN is the value of its root.

SPNs allow for time linear computation of conditional probabilities, among other inference computations, by means of a bottom-up pass from the indicators to the root. An SPN is therefore a function of the indicator variables $S(x_1, ..., x_n, \overline{x_1}, ..., \overline{x_n})$. Evaluating a SPN for a given configuration of the indicator $\lambda = (x_1, ..., x_n, \overline{x}_1, ..., \overline{x}_n)$ is done by propagating from the leaves to the root. When all indicators are set to 1 then $S(\lambda)$ is the partition function of the unnormalised probability distribution. (If $S(\lambda) = 1$ then the distribution is normalised.) The scope of an SPN is defined as the set of variables appearing in it. An essential property of SPNs as a deep architecture with tractable inference is the one of validity [Poon and Domingos, 2011]: intuitively, if an SPN is valid, its expansion includes all monomials present in its network polynomial [Darwiche, 2003]. (Cf. Appendix for an example.)

### 2.2 Credal Sum-product Networks (CSPNs)

In recent work [Mauá et al., 2017], so-called credal sum-product networks (CSPNs) were proposed, which are a class of imprecise probability models that unify the credal network semantics with SPNs. That work was particularly aimed at analysing the robustness of classification in SPNs learned from data: first they consider a standard SPN [Gens and Domingos, 2013], and from that they obtain the CSPN by an "$\epsilon$-contamination" of the SPN (cf. [Mauá et al., 2017]).

A CSPN is defined in the same way as a SPN except that it allow weights on sum nodes to vary in a closed and convex set, which then banks on the notion of a **credal set**.

**Definition 3.** [Credal Sum-Product network] A CSPN over variables $X_1, \ldots, X_n$ is a rooted directed acyclic graph whose leaves are the indicators $(x_1, \ldots, x_n, \overline{x}_1, \ldots, \overline{x}_n)$ and whose internal nodes are sums and products. Each sum node $i$ is associated to a credal set $K_i$. A credal set is a convex set of probability distributions.

A credal set can be interpreted as a set of imprecise beliefs meaning that the true probability measure is in that set but due to lack of information, that cannot be determined. In order to characterize a credal set,

---

2. For reasons of space, most of the examples as well as expanded notes on some of the concepts raised in the paper can be found in the Appendix.

one can use a (finite) set of extreme points (edges of the polytope representing the credal set), probability intervals or linear constraints.[3] (Cf. Appendix for an example.)

Although [Mauá et al., 2017] do not consider the notion of validity for their structures, we can update the existing formulation and define a CSPN to be *valid* if it satisfies the same conditions as would valid SPNs, namely *consistency* and *completeness*.

Inference in CSPN boils down to the computation of minimum and maximum values for $\lambda$, that is, a given configuration of the indicators. This amounts to computing the lower and upper likelihood of evidence, which can be performed in almost the same fashion as for SPNs. More precisely, we evaluate a CSPN from leaves toward the root. The value of a product node is the product of the values of its children (as in SPNs). The value of a sum node is given by optimising (either maximising or minimising) the following equation: $\sum_{j \in Ch(i)} w_{ij} * v_j$ where the weights $w_{ij}$ are chosen from the credal set $\mathbf{w}_i$. (Here, weights are chosen subject to constraints that define a probability distribution within the credal set.) The value of a CSPN, denoted $S(x_1, \ldots, x_n, \overline{x_1}, \ldots, \overline{x_n})$, is the value of its root node.[4]

In particular, [Mauá et al., 2017] provide ways to compute the minimum and maximum values for a given configuration $\lambda$ of the indicator variables, as well as ways to compute conditional expectations by means of a linear program solver to solve (maximise or minimise) and find the best weight to propagate. But that can be challenging: since we are interested in learning SPN structures, such inference tasks would need to be solved in each iteration. Thus, we argue that reasoning with extreme points of the credal set turns out to be more practical and efficient. In the interest of space, the Appendix includes a brief exposition on this idea.

## 3. Learning a Credal Sum-product Network

Algorithms to learn sum-product networks are maturing; we can broadly categorise them in terms of discriminative training and generative learning. Discriminative training [Gens and Domingos, 2012] learns conditional probability distributions while generative learning [Gens and Domingos, 2013] learn a joint probability distribution. In the latter paradigm, [Gens and Domingos, 2013] propose the algorithm Learn-SPN that starts with a single node representing the entire dataset, and recursively adds product and sum nodes that divide the dataset into smaller datasets until a stopping criterion is met. Product nodes are created using group-wise independence tests, while sum nodes are created performing clustering on the row instances. The weights associated with sum nodes are learned as the proportion of instances assigned to a cluster.

In this paper, we are tackling learning credal sum-product networks from missing values. In the sequel, in order to illustrate the different parts of the algorithm, we will refer to the following table containing a dataset with missing values. More precisely, let us consider a dataset that contains 600 instances on 7 (Boolean) variables. We consider the particular case of missing data on two variables $A$ and $C$ with 1% of missing data depicted in Table 1.

In this work, we propose a generative learning method to learn a CSPN from missing values. In the same fashion as LearnSPN does, we start from a single node and recursively add sum or product nodes. The next two subsections detail the process and equations related to adding a product or sum node.

---

3. It is important to note that the number of extreme points can reach $N!$ where $N$ is the number of interpretations [Wallner, 2007].

4. We remark that in this paper for simplicity, we restrict ourselves to Boolean variables, but this is easily generalised to many-valued variables.

Table 1: Dataset example

| instance number | A | B | C | D | E | F | G |
|---|---|---|---|---|---|---|---|
| 1 | 1 | 1 | 1 | 1 | 1 | 1 | 1 |
| 2 | ? | 1 | ? | 0 | 0 | 1 | 1 |
| 3 | ? | 1 | ? | 0 | 0 | 0 | 1 |
| 4 | ? | 0 | 1 | 0 | 0 | 1 | 0 |
| 5 | ? | 0 | ? | 1 | 0 | 0 | 0 |
| 6 | ? | 1 | ? | 0 | 1 | 1 | 1 |
| 7 | ? | 1 | ? | 0 | 0 | 0 | 1 |
| 8 | 0 | 1 | 1 | 1 | 1 | 1 | 1 |
| ... | | | | | | | |

## 3.1 Group-wise Independence Test

In the first loop of LearnCSPN algorithm (see Algorithm 1), we are first looking for a set of independent variables, and do so using a group-wise independence test. This consists in computing a G-value. This G-value is compared to a threshold. The lower the G-value is the more likely chance there is that the two variables are independent.

Formally, $G(X_1, X_2) = 2\sum_{x_1}\sum_{x_2} m(\underline{C}, \overline{C})(x_1, x_2) * \log \dfrac{m(\underline{C}, \overline{C})(x_1, x_2) * |T|}{m(\underline{C}, \overline{C})(x_1) * m(\underline{C}, \overline{C})(x_2)}$, where $m$ is the mean function, $\underline{C}$ and $\overline{C}$ are count functions that are detailed below.

We consider two count functions, that we denote $\underline{C}$ and $\overline{C}$. The first function takes into account only instances where the values of the variables involved in the independence test are complete. More precisely, it counts the occurrences (number of instances) matching the values $x_1, x_2$ (analogously, $\underline{C}(x_1)$ counts the occurrences of $x_1$). The second function $\overline{C}$ takes into account all instances. More precisely, $\overline{C}(x_1)$ counts the number of instances matching $x_1$ and adding to that every instance where $X =?$ is present. Formally, $\overline{C}$ is as follows: $\overline{C}(x_1, x_2) = |insts \models (x_1, x_2)| + |insts \models (?, x_2)| + |insts \models (x_1, ?)|$

Intuitively, $\overline{C}$ acts as a gatekeeper that takes into account missing information and readjusts the $G$-value accordingly. The function we use to balance the two count functions is the mean, it allows us to not prioritise one case above the other. A more elaborate measure other than the mean could be applied too, of course; for example, we could take into account the proportion of missing values and define a measure that is based on that proportion.

Let us consider the counts for (A,B) and (A,C) in Table 2. Every value of B is available. In contrast, for A and C we have some missing values. We can compute the $G$-value for the two functions $\underline{C}$ and $\overline{C}$.

Table 2: Counts for the tuples (A,B) and (A,C)

| | | | | | | | |
|---|---|---|---|---|---|---|---|
| $\underline{C}(a,b)$ | 210 | $\overline{C}(a,b)$ | 214 | $\underline{C}(a)$ | 242 | $\overline{C}(a)$ | 248 |
| $\underline{C}(a,\bar{b})$ | 32 | $\overline{C}(a,\bar{b})$ | 34 | $\underline{C}(\bar{a})$ | 352 | $\overline{C}(\bar{a})$ | 358 |
| $\underline{C}(\bar{a},b)$ | 304 | $\overline{C}(\bar{a},b)$ | 308 | $\underline{C}(b)$ | 514 | $\overline{C}(b)$ | 522 |
| $\underline{C}(\bar{a},\bar{b})$ | 48 | $\overline{C}(\bar{a},\bar{b})$ | 50 | $\underline{C}(\bar{b})$ | 80 | $\overline{C}(\bar{b})$ | 84 |
| $\underline{C}(a,c)$ | 150 | $\overline{C}(a,c)$ | 156 | $\underline{C}(c)$ | 200 | $\overline{C}(c)$ | 212 |
| $\underline{C}(a,\bar{c})$ | 92 | $\overline{C}(a,\bar{c})$ | 97 | | | | |
| $\underline{C}(\bar{a},c)$ | 50 | $\overline{C}(\bar{a},c)$ | 56 | $\underline{C}(\bar{c})$ | 394 | $\overline{C}(\bar{c})$ | 404 |
| $\underline{C}(\bar{a},\bar{c})$ | 302 | $\overline{C}(\bar{a},\bar{c})$ | 307 | | | | |

Here, $G(A, B) = 0.00457$ for $\underline{C}$ and $G(A, B) = -2.61699$ for $\overline{C}$. As for (A,C) we have $G(A, C) = 33.3912$ for $\underline{C}$ and $G(A, B) = 28.6183$ for $\overline{C}$ which implies that in both cases A and B are independent and A and C are dependent.

## 3.2 Clustering Instances

When no independent set of variables is found, we perform hard EM to cluster the instances. Intuitively, a cluster is a set of instances where variables mostly have the same values. However, when dealing with missing values, it is less obvious how one is to evaluate an instance to a cluster. To that end, we will be using two measures of likelihood. Let us motivate that using an example: consider the following cluster $cl1$ which contains 40 instances including 36 complete instances and the cluster $cl2$ which contains 70 instances including 51 complete instances. Statistics of $cl1$ and $cl2$ are shown in Table 3.[5]

Table 3: Statistics for clusters $cl1$ and $cl2$

| $cl1$ | A | B | C |
|---|---|---|---|
| 0 | 40 | 30 | 5 |
| 1 | 0 | 6 | 33 |

| $cl2$ | A | B | C |
|---|---|---|---|
| 0 | 60 | 6 | 61 |
| 1 | 10 | 45 | 9 |

When considering a complete instance (*e.g.* 0 0 1), we only use one measure which we call the lower measure of log-likelihood, denoted $\underline{\log}_{cl}(inst)$, which only takes into account the statistics of the known values as shown in Table 3. More precisely, this is expressed as:[6]

$$\underline{\log}_{cl}(inst) = \sum_{var} \log((w_{val} + smoo)/(nb\_inst + 2 * smoo)) \tag{1}$$

where $w_{val}$ is the number of instances where the value for the variable *var* is also the value of the instance, *val*. $nb\_inst$ is the number of instances where the value for variable *var* is known. For instance, given the complete instance $inst\_1 = 0\ 0\ 1$, we have $\underline{\log}_{cl1}(001) = -0.1434$ and $\underline{\log}_{cl2}(001) = -1.8787$.

If we consider the incomplete instance $inst\_2 = 0\ 0\ ?$, we have $\underline{\log}_{cl1}(00?) = -0.0812$ and $\underline{\log}_{cl2}(00?) = -0.9914$. Note that statistics on incomplete instances for B and $\overline{C}$ are not used and yet it is clear that $cl1$ is the best cluster. This motivates a second measure of log-likelihood, that we call upper log-likelihood $\overline{\log}_{cl}(inst)$. This measure takes into account the lower measure to which we had a log value of the worst case scenario:

$$\widetilde{\log}_{cl}(inst) = \sum_{var} \log((w + smoo)/(nb\_inst + 2 * smoo)) - \log((inc + smoo)/(all\_inc + 2 * smoo)) \tag{2}$$

where $w$ is the number of instances where the value for the variable *var* is that value that is poorest fit. The term *inc* corresponds to the number of instances in the cluster where the value is unknown but has been assigned to the worst case scenario when added to the cluster. Then, *all_inc* is the number of instances where the value is unknown.

Formally,

$$\overline{\log}_{cl}(inst) = \widetilde{\log}_{cl}(inst) + \underline{\log}_{cl}(inst). \tag{3}$$

---

5. Missing values are only present for variables B and C.
6. $2 * smoo$ is the smoothing value multiplied with the number of possible values of *var*; since we consider a Boolean dataset, it is 2.

Intuitively, we take into account all instances for the computation of the upper log-likelihood, and consider incomplete instances in a manner that does not exaggerate the uncertainty relative to the complete instances. In our example, with *inst_2* 0 0 ?, we therefore have $\overline{\log}_{cl1}(00?) = -0.9355$ and $\overline{\log}_{cl2}(00?) = -1.8787$.

Specifically, the lower bound refers to the SPN that would be learned if we only consider the complete instances. And the upper bound denotes the learning regime if every incomplete instance has been made complete in a way that fits the cluster. The overall algorithm that clusters the set of instances is given in Algorithm 3 in Appendix. Intuitively, by computing these two measures, we allow an instance to appear in more than one cluster. For an instance, we compute the value of the *best* lower and upper log-likelihood, and the instance therefore belongs to every cluster for which the lower log-likelihood is better than the *best* upper log-likelihood.

Now that all instances have been clustered, we create a sum node with as many edges as clusters found. And to compute the weights associated to each edge, we are looking at the proportion of instances in each cluster. As explained in the clustering process, we allow an instance to belong to multiple clusters if this instance is incomplete. This implies that the intersection between clusters might not be empty, which facilitate the construction of probability interval for each edge. We compute the lower bound of the interval with the proportion of complete instances in the cluster and the upper bound with the proportion containing both complete and incomplete instances in the cluster (Cf. Appendix for examples).

The overall algorithm to learn a credal SPN is given in Algorithm 1.

---

**Algorithm 1** LearnCSPN(T, V)

---

**Require:** set of instances T and set of variables V
**Ensure:** a CSPN representing an imprecise probability distribution over V learned from T
 1: **if** |V| = 1 **then**
 2:     **return** univariate imprecise distribution estimated from the variable's values in T
 3: **else**
 4:     partition V into approximately independent subsets $V_j$
 5:     **if** success **then**
 6:         **return** $\prod_j$ LearnCSPN(T, $V_j$)
 7:     **else**
 8:         partition T into subsets of similar instances $T_i$
 9:         **return** $\sum_i K_i$ * LearnCSPN($T_i$, V)
10:     **end if**
11: **end if**

---

### 3.3 Properties of LearnCSPN

LearnCSPN of Algorithm 1 holds some interesting properties as stated in the following.

**Theorem 4.** *LearnCSPN learns a complete and consistent CSPN, and thus, a valid CSPN.*

LearnCSPN holds a second and fundamental property with respect to the downward compatibility of CSPNs with SPNs. Indeed, when learning a CSPN from a complete dataset, the weights are no longer intervals but single degrees: when there is no missing value in the dataset, the count function for the independence test matches the count function of LearnSPN. Analogously, the computation of the log-likelihood of an instance in a cluster corresponds to the computation of the lower log-likelihood.

**Theorem 5.** *LearnCSPN on a complete dataset is equivalent to LearnSPN.*

It can also be shown that the inference regime is tractable (cf Appendix): basically, LearnCSPN learns a CSPN where internal nodes have at most one parent, and it is known [Mauá et al., 2017] that computing lower/upper conditional expectations of a variable in CSPNs takes at most polynomial time when each internal node has at most one parent.

## 4. Experiments

We evaluate our algorithm to learn a credal sum-product network in two ways, by evaluating the accuracy of the learned model in terms of log-likelihood and the accuracy of the model when performing inference. In particular, we compare learnCSPN to the classical LearnSPN algorithm [Gens and Domingos, 2013] which learns a SPN given full datasets (*i.e.* no missing values).[7] To that end, we have applied LearnCSPN (resp. learnSPN) on various datasets drawn from [Gens and Domingos, 2013].[8]

Learning from missing values widely use the Expectation-Maximization (EM) procedure. In that sense, it is reasonable to compare the two learning regimes (LearnSPN and LearnCSPN) together as the models learned are from the same category of tractable graphical models, which makes it easier to relate. For the future, it will be worth comparing LearnCSPN to different models learned with the EM procedure to deal with missing values.[9]

### 4.1 Settings

For our purposes, to study the effect of missing values, we consider two settings: 1% and 5% of missing values. As the datasets obtained are complete and since the comparison is done with learning a SPN on complete data, we have to remove values to instantiate a setting for learning the CSPN. We simply modify the existing datasets by removing values from instances on 1% and 5% of the set of instances.[10] The number of values removed on each instance can vary and is randomly chosen. (A degenerate case with all missing values on an instance may happen, but we do not explicitly test for it.)

To better illustrate the relevance of our results, we have computed three different log-likelihood scores based on the learned CSPN. These are:

- A minimal log-likelihood is given by considering the worst-case scenario, *i.e.* when we select a compatible SPN with the weights that computes the worst log-likelihood score.

- An average log-likelihood is given by a compatible SPN with weights that gives the average log-likelihood-score.

---

7. http://spn.cs.washington.edu/learnspn/
8. Characteristics of the datasets can be found in Appendix, Table 6.
9. It is important to note a major distinction in our comparisons. Since SPNs appeal to EM to handle missing values, this allows a reasonable comparison, but even then, there is principle difference. Recall that CSPNs allows the user to recognise that there is incomplete knowledge and uncertainty about predictions as characterized by imprecise probabilities. Indeed, as argued earlier, there are significant benefits to using credal and other imprecise probability representations that are not directly assessed in standard structure learning evaluations. Likewise, there has been considerable work on clustering with incomplete data [Wagstaff, 2004]. But the objective in our work is not to simply produce the best clusters, but rather to capture the uncertainty present in the data so as to model random variables with imprecise probabilities. Thus, drawing comparisons to such clustering approaches is not direct or obvious. More generally, to have a deeper understanding of the wider applicability of CSPNs, we need a more comprehensive analysis of their use in a wide variety of applications, together with a reasonable means to draw comparisons with related efforts. We hope this work serves as a useful starting point for such investigations.
10. That is, 99% and 95% of the instances are complete instances. Note also that missing values only appear on the training set.

- Finally, an optimised log-likelihood that is given by considering the set of extreme points that maximises the log-likelihood score. More precisely, for each test case, we choose the compatible SPN that maximises the log-likelihood, the overall optimised value is the average over the test set.

## 4.2 Learning Credal SPNs

As discussed, we compare LearnSPN and LearnCSPN. On the former, the heuristics used are hard incremental EM over a naive Bayes mixture model to cluster the instances and a G-test of pairwise independence for the variable splits. The heuristics for LearnCSPN are the one given in Section 3. In both algorithms, we ran ten restarts through the subset of instances four times in random order. Table 4 depicts the three score as defined in the settings, the time of the learning computation (in s), and the size (in terms of number of nodes) of the learned SPNs and CSPNs.

The results showed in **bold** relate to the significant improvement whether in terms of size of the model or in terms of log-likelihood scores.

Table 4: Log-likelihood given missing values

| Dataset | Log Likelihood + **time** (s)/ **size** (number of nodes) | | | | | | |
| | 0% | missing values on **1%** of inst | | | missing values on **5%** of inst | | |
| | LearnSPN | avg_ll | min_ll | opt_ll | avg_ll | min_ll | opt_ll |
|---|---|---|---|---|---|---|---|
| NLTCS | -6.114 | -6.219 | -6.229 | **-6.092** | -5.466 | -5.504 | **-5.272** |
| | 47.981 / 2584 | 932.832 / 53811 | | | 891.506 / 5458 | | |
| Plants | **-12.988** | -13.179 | -13.190 | -13.028 | -14.909 | -14.956 | -14.541 |
| | 137.057 / 20714 | 662.244 / **8661** | | | 4387.299 / **14281** | | |
| Audio | -40.528 | -34.211 | -34.264 | **-33.811** | -33.755 | -33.927 | **-33.129** |
| | 770.684 / 35491 | 7307.784 / 72822 | | | 1231.921 / **2021** | | |
| Jester | -53.444 | -48.666 | -48.685 | **-48.299** | -48.288 | -48.338 | **-47.964** |
| | 1082.599 / 47769 | 4143.19 / 87467 | | | 13199.7 / 101708 | | |
| Netflix | -57.408 | -53.569 | -53.579 | **-53.326** | -53.886 | -53.917 | **-53.484** |
| | 1145.967 / 39447 | 5682.157 / 43229 | | | 16367.691 / 37674 | | |
| Retail | -11.131 | -8.173 | -8.183 | **-8.114** | -6.364 | -6.410 | **-6.260** |
| | 83.637 / 3790 | 1631.054 / 6335 | | | 5701.983 / 6937 | | |
| Pumsb-star | **-25.548** | -30.219 | -30.229 | -30.197 | -30.970 | -31.014 | -30.787 |
| | 431.414 / 73949 | 4154.584 / 347880 | | | 1606.189 / **2128** | | |
| DNA | -85.272 | -77.601 | -77.639 | **-77.174** | -82.261 | -82.475 | **-81.192** |
| | 66.37 / 22627 | 220.202 / 18957 | | | 146.218 / 5069 | | |
| MSWeb | -10.212 | -7.421 | -7.430 | **-7.344** | -6.921 | -6.963 | **-6.753** |
| | 175.64 / 11000 | 12035.587 / 9145 | | | 26367.275 / 3836 | | |
| Book | -36.656 | -21.571 | -21.592 | **-21.441** | -17.484 | -17.568 | **-17.323** |
| | 304.339 / 95077 | 5245.763 / **46970** | | | 17944.24 / **68137** | | |
| EachMovie | -52.836 | -44.023 | -44.058 | **-43.548** | -42.239 | -42.400 | **-41.812** |
| | 229.943 / 86066 | 1732.422 / **16808** | | | 4508.381 / **31141** | | |
| WebKB | -158.696 | -132.869 | -132.921 | **-132.033** | -110.392 | -110.595 | **-109.624** |
| | 493.989 / 241777 | 2518.345 / **21075** | | | 3339.987 / **22066** | | |
| Reuters-52 | -85.995 | -73.123 | -73.134 | **-73.106** | -73.308 | -73.366 | **-73.259** |
| | 1507.077 / 427220 | 11128.818 / **22511** | | | 18168.615 / **4768** | | |
| 20 Newsgrp. | -159.701 | -95.002 | -95.101 | **-94.465** | -92.998 | -93.518 | **-92.271** |
| | 20656.808 / 3047296 | 28115.16 / **154923** | | | 122947.065 / **197478** | | |
| BBC | -248.931 | -196.850 | -197.482 | **-195.062** | -179.487 | -179.751 | **-197.397** |
| | 883.801 / 290423 | 2623.016 / **62194** | | | 1824.548 / **9976** | | |
| Ad | **-27.298** | -28.545 | -28.606 | -28.097 | -31.452 | -31.498 | -31.267 |
| | 10197.343 / 690272 | 9436.839 / **59582** | | | 9383.391 / **9772** | | |

From Table 4, it is clear that in most datasets (13 out of 16), the best log-likelihood is given by the optimised log-likelihood computed on the learned CPSN. And this is observed regardless of whether we consider a small amount of missing data (1%) or a slightly larger amount (5%). For datasets where learnSPN gives the best log-likelihood, *e.g.* Plants and Ad, we notice that the number of nodes in the learned CSPN is much lesser than the learned SPN. (Also note that the difference between log-likelihood values in that case is not significant.)

One drawback of LearnCSPN is the computational time during learning. As explained in Section 3, it is due to the consideration of an instance in multiple clusters as well as the computation of more values during the clustering process. This can be improved by considering more efficient data structures to store the clustering information, which would make for interesting future work. We will now report on the efficiency of LearnCSPN with inference.

### 4.3 Inference in Credal SPNs

In this section, we evaluate the accuracy of the learned model at query time. This amounts to computing the probability of a set of query variables given evidence. Results given in plots report the average conditional log-likelihood (CLL) of the queries with respect to the evidence. As discussed in [Gens and Domingos, 2013], it is sufficient to report the CLL since it approximates the KL-divergence between the inferred probability and the true probability.

For the sake of this second evaluation, we generate 1000 queries (for each dataset) from the test set, varying the proportion of query variables in {.10, .20, .30, .40, .50} and fixing the proportion of evidence variables to .30. And in a second stage, we fix the proportion of query variables to .30 and vary the proportion of evidence variables in {0, .10, .20, .30, .40, .50}. We report the results in plot graphs for datasets **book** and **20 Newsgrp** (*cf.* Figure 1).[11] We only present the computation of the optimised conditional log-likelihood given its general better accuracy in our prior experimental setup. From Figure 1, it is clear that the positive results from Table 4 are reaffirmed by the results of inference accuracy; overall, learned CSPNs offer high prediction accuracy.

The experimental results reveal two majors advantages:

- by allowing weights on sum nodes to vary in a credal set (specified by interval or extreme points), we allow for a model that is more compact and has better accuracy. Indeed, a CSPN can be seen as a set of compatible SPNs, each SPN being drawn from a completion of the missing values.

- as we see in Table 4, even the worst case scenario gives better accuracy on 13 datasets. This means that one can learn a better SPN from missing values by first learning a CSPN and then extracting a compatible SPN (perhaps even a random compatible SPN).

### 5. Conclusions

In this work, we were motivated by the need for a principled solution for learning probabilistic models in the presence of missing information. Rather than making a closed-world/complete knowledge assumption that might ignore or discard the corresponding entries, which is problematic, we argued for learning models that natively handle imprecise knowledge. Consequently, by leveraging the credal network semantics and the recent paradigm of tractable learning, we proposed and developed a learning regime for credalSPNs. Our empirical results show that the regime performs very strongly on our datasets, not only in terms of the log-likelihood and the size of the resulting network, but also in terms of prediction accuracy.

---

11. The plot graphs for the remaining datasets can be found in the Appendix.

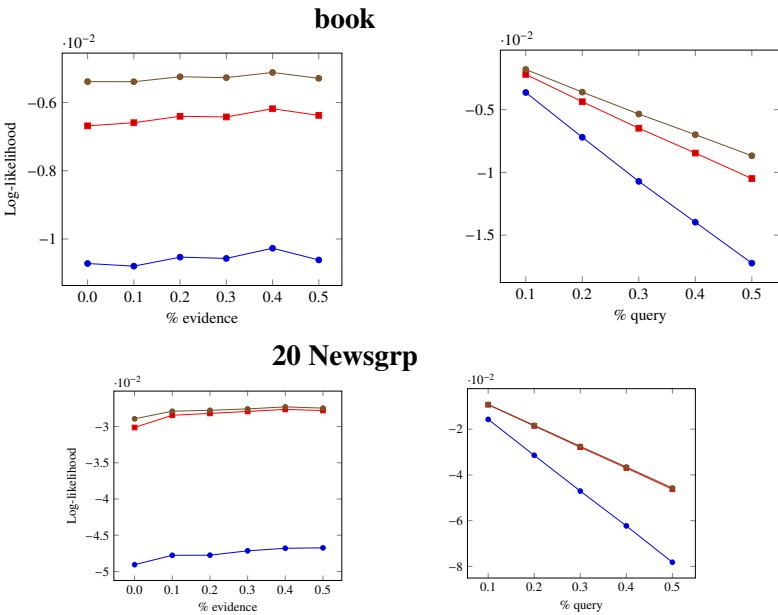

Figure 1: Average conditional log-likelihood normalised by the number of query variables, where blue dots denote LearnSPN, red squares denote LearnCSPN with 1% missing data and brown dots denote LearnCSPN with 5% missing data. Left plot fixes the fraction of evidence variables at 30% and varies the fraction of query variables; Right plot fixes query variables at 30% and varies evidence.

Directions for future work include pushing the applicability of LearnCSPNs on a large-scale text data applications, while possibly considering relational extensions [Nath and Domingos, 2015] to leverage extracted and mined relations that may be obtained by natural language processing techniques. For the immediate future, we are interested in optimising the algorithm for minimising the discarded convex regions when extracting extreme points.

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

## 6. Appendix

### 6.1 SPN Validity & Example

An SPN is valid if it satisfies the two following conditions:

1. an SPN is *complete*: if and only if all children of the same sum node have the same scope.

2. an SPN is *consistent*: if and only if no variable appears negated in one child of a product node and non-negated in another.

As mentioned, SPNs are essentially based on the notion of network polynomials [Darwiche, 2003]; an SPN can be expanded to a network polynomial as seen in Example 6. Intuitively, if an SPN is valid, its expansion includes all monomials present in its network polynomial.

**Example 6.** Figure 2 illustrates an example of a Sum-Product network over two Boolean variables.

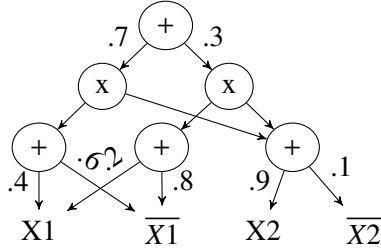

Figure 2: Sum-Product network example

The function $S$ of the indicators is written as

$$S\left(x_1, x_2, \overline{x_1}, \overline{x_2}\right) = \ .7\left(.4x_1 + .6\overline{x_1}\right)\left(.9x_2 + .1\overline{x_2}\right) + \\ .3\left(.2x_1 + .8\overline{x_1}\right)\left(.9x_2 + .1\overline{x_2}\right)$$

Its expansion into a network polynomial is given by

$$(.7 * .4 * .9 + .3 * .2 * .9)x_1x_2 + (.7 * .4 * .1 + .3 * .2 * .1)x_1\overline{x_2} + \\ (.7 * .6 * .9 + .3 * .8 * .9)\overline{x_1}x_2 + (.7 * .6 * .1 + .3 * .8 * .1)\overline{x_1}\overline{x_2} \\ = .306x_1x_2 + .034x_1\overline{x_2} + .594\overline{x_1}x_2 + .066\overline{x_1}\overline{x_2}$$

It is worth noting also that the SPN in Figure 2 is valid.

### 6.2 Credal SPN Example

**Example 7.** We illustrate an example of a CSPN in Figure 3. In this example, along the lines of the discussions in [Mauá et al., 2017], the weights are given as a convex set of different types. That is, weights $w_1, w_2$ are probability intervals, as are weights $w_5, w_6$. In contrast, $w_3, w_4$ and $w_7, w_8$ are sets of extreme points (*i.e.* described as the convex hull by the set of extreme points). We have to ensure that the constraints associated with probability intervals realise a normalised distribution.

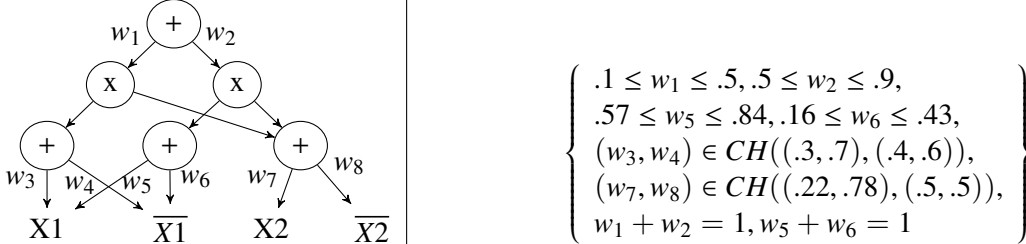

Figure 3: Credal Sum-Product network example

## 6.3 Inference in Credal SPNs

As mentioned, [Mauá et al., 2017] provide ways to compute the minimum and maximum values for a given configuration $\lambda$ of the indicator variables, as well as ways to compute conditional expectations, which we describe in the sequel. In particular, they propose to use a linear program solver to solve (maximise or minimise) and find the best weight to propagate. We explain later that it is easier to learn intervals of probability degrees when clustering. But we also show here that inference using extreme points can be more efficient. Thus, we recall the definition of interval-based probability distribution and an inference scheme using a linear program. The second part of this section tackles the extraction of extreme points from the interval-based probability distribution.

### 6.3.1 INFERENCE USING A LINEAR PROGRAM SOLVER

Interval-based probability distributions (IPD for short) are a very natural and common way to specify imprecise and incomplete information. In an IPD $IP$, every interpretation $\omega_i \in \Omega$ is associated with a probability interval $IP(\omega_i) = [l_i, u_i]$ where $l_i$ (resp. $u_i$) denotes the lower (resp. upper) bound of the probability of $\omega_i$.

**Definition 8.** [Interval-based probability distribution] Let $\Omega$ be the set of possible worlds. An interval-based probability distribution $IP$ is a function that maps every interpretation $\omega_i \in \Omega$ to a closed interval $[l_i, u_i] \subseteq [0, 1]$.

Given a credal set, if the weights are depicted as extreme points, we can obtain an IPD by taking into account for each value the minimum degree of the extreme points (for the lower bound) and the maximum degree of the extreme points (for the upper bound). The result, as any other interval-based probability distribution, should satisfy the following constraints in order to ensure that the underlying credal set is not empty and every lower/upper probability bound is reachable.

$$\sum_{\omega_i \in \Omega} l_i \leq 1 \leq \sum_{\omega_i \in \Omega} u_i \tag{4}$$

$$\forall \omega_i \in \Omega, \; l_i + \sum_{\omega_{j \neq i} \in \Omega} u_j \geq 1 \; \text{ and } u_i + \sum_{\omega_{j \neq i} \in \Omega} l_j \leq 1 \tag{5}$$

Let us illustrate an interval-based probability distribution in the context of a credal SPN.

**Example 9.** Let $S$ be a sum node with three children $D_S = \{cl_1, cl_2, cl_3\}$. Table 5 provides an example of interval-based probability distribution.

This imprecise probability distribution satisfies the two conditions of Equations (4) and (5).

Table 5: Example of an interval-based probability distribution over a set of 3 clusters.

| $S$ | $IP(S)$ |
|-----|---------|
| $cl_1$ | [ 0, .4 ] |
| $cl_2$ | [.1, .55] |
| $cl_3$ | [.1, .65] |

From intervals, as depicted in Table 5, it is easy to compute, using a linear program solver, $\min_w S_w(\lambda)$ and $\max_w S_w(\lambda)$ as seen in the following example.

**Example 10.** [Continued] Given Table 5, let us consider that the weights associated to the children of the clusters have been optimised to .4 for cluster 1, .5 for cluster 2 and .1 for cluster 3. Thus, we encode the linear program as follow:

```
Maximize
        value:  .4*cl1  +  .5*cl2  +  .1*cl3

Subject To
        c0:       cl1  +  cl2  +  cl3  =  1

Bounds
        0  <=  cl1  <=  0.4
        0.1  <=  cl2  <=  0.55
        0.1  <=  cl3  <=  0.65

End
```

### 6.3.2 REASONING ABOUT EXTREME POINTS

We present a simple algorithm below that can extract a set of extreme points. This set is not the complete set as the number of extreme points for $n$ values can reach $n!$ extreme points [Wallner, 2007].

An example will be given later but most significantly, inference no longer requires a call to a linear program solver to compute the maximum or minimum value. It suffices to browse the set of extreme points and find the point which gives the maximum or minimum value. While the complexity of inference stays the same, it significantly simplifies the computational efficiency of the regime as a whole, as the extraction is made during learning.

## 6.4 Examples on Weights in SPNs

**Example 11.** Let us consider Figure 4 which depicts a small credal network over 3 variables and a dataset where each row is an instance that represents a record at time $t$ of the variables (or perhaps the record from an expert about the variables). Some of these instances are incomplete, denoted with a '?.'

Since the structure is given, we can easily learn the imprecise probability distributions associated to nodes $A, B$ and $C$. The lower bound of an interpretation is given by the proportion of **known instances** compatible with the interpretation. For example, $\underline{IP}(A = 1) = 1/2$. As for the upper bound, the degree

**Algorithm 2** Extract a set of extreme points for a sum node with $n$ children (Function called **Extract**)

---

**Require:** *llambda*    ▷ Initialize as the sum of the lower weights associated to each edge of the sum node

**Require:** *PT* be an array representing an extreme point ▷ Initialize as the lower weights associated to each edges of the sum node

**Require:** *expl* the list of index of weight of *PT* already fixed

**Ensure:** A set of extreme points *ext_points*

  1: **for** index $i$ in $1, \ldots n$ **do**

  2:     **if** $i$ is not in *expl* **then**                                        ▷ $PT[i]$ has not been fixed

  3:         **if** $llambda <= (Wupp[i] - Wlow[i])$ **then**

  4:             $temp = PT[i]$

  5:             $PT[i] + = llambda$

                                                    ▷ A extreme point *PT* is found

  6:             **if** $PT$ not in *ext_points* **then**

  7:                 *ext_points.add*$(PT)$

  8:             **end if**

  9:             $PT[i] = temp$

10:         **else**

11:             $temp = PT[i]$

12:             $PT[i] = Wupp[i]$

13:             $expl.add(i)$

14:             **Extract**$(PT, llambda - Wupp[i] + Wlow[i]$ ,*expl*$)$

15:             $PT[i] = temp$

16:         **end if**

17:     **end if**

18: **end for**

---

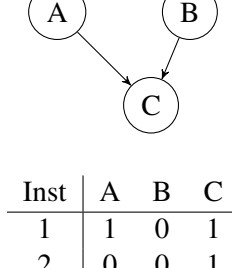

| Inst | A | B | C |
|------|---|---|---|
| 1 | 1 | 0 | 1 |
| 2 | 0 | 0 | 1 |
| 3 | 1 | ? | 1 |
| 4 | ? | 1 | 0 |

Figure 4: Credal network structure and a corresponding dataset

is given by the proportion of instances that *are or could* potentially (owing to the missing information) be compatible with the interpretation. For example, $\overline{I\!P}(A = 1) = 3/4$.

Likewise, the lower (respectively, upper) bound for a cluster is given by the number of complete instances (respectively, the number of incomplete instances). Unfortunately, this method does not ensure that all interval bounds are reachable. Let us illustrate such an example.

**Example 12.** Let us consider a partition of 80 instances, which contains 30 incomplete instances (and so, 50 complete). Let us say that the partition has been clustered into 5 clusters. One extreme case could be that the first cluster contains 10 complete instances and 30 incomplete. The other clusters split the remaining complete instances. This means that the lower and upper bound of the interval associated with the second, third and forth cluster are the same. Only the first cluster has different bounds, indeed $I\!P(c1) = [.125, .5]$. Yet, the probability of the three other clusters add up to .5 which implies that .125 cannot be reached.

A simple and effective way to deal with this problem is to readjust the bound to be able to satisfy the conditions of a well-defined interval-based probability distribution (as given by Equations (4) and (5)). More precisely, from the learned interval, we extract a set of extreme points that defines a convex set included in the actual convex set determined by the intervals (cf. Example 13). Note that the extraction process does not extract all the extreme points. This is purposeful and is motivated by two reasons:

- the number of extreme points can be extremely high (at most $n!$) [Wallner, 2007];

- between two extreme points, the change in the probability degrees might be small which would not gravely affect the overall log-likelihood or conditional probability.

**Example 13.** In this example, we illustrate the convex set covered in the process of extraction of extreme points. Note that in case of a small amount of clusters, the two reasons discussed above do not manifest so strongly. Let us consider a sum node with 3 clusters with the following interval weights:

- c1 [.2,.4]

- c2 [.3,.55]

- c3 [.2,.48]

There are 6 extreme points as shown below:

- P1 [0.4, 0.4, 0.2]

- P2 [0.4, 0.3, 0.3]

- P3 [0.25, 0.55, 0.2]

- P4 [0.2, 0.55, 0.25]

- P5 [0.22, 0.3, 0.48]

- P6 [0.2, 0.32, 0.48]

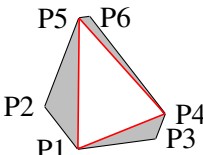

Figure 5: Convex set of extreme points extracted

In this example, it is easy to extract all extreme points since it only involves 3 clusters. However, on a huge dataset we might obtain (say) a hundred clusters. In this case, the number of extreme points can reach 100!. To avoid the computation of all the extreme points and also avoid having to consider all of them in the computation of the log-likelihood, we restrict ourselves to a subset of the extreme points which defines a sub convex set as illustrated in Figure 5. In our example, this results in considering 3 extreme points over the 6 possible ones, and the grey areas are the convex sets of degrees that are ignored.

Nonetheless, the data structure used to store the extreme points can be easily optimised for also storing the maximum weight which can augment the efficiency of inference. Analogously, the set of extreme points we select can be improved to minimise the convex regions that are ignored.

## 6.5 Clustering Algorithm

Algorithm 3 given below summarises how the set of clusters is build, this is done by computing the log-likelihood measures defined in Section 3.

## 6.6 Complexity Remarks

The previous result stated that LearnCSPN returns a valid CSPN. We can now be precise in our analysis of the learned CSPN, and its structure. In fact, if we remove leaves then the CSPN is a tree-shaped network. This is formally stated in the next proposition.

**Proposition 14.** *LearnCSPN learns a CSPN where internal nodes have at most one parent.*

This proposition allows us to leverage a strong result on the polynomial time computational complexity for conditional expectations, as proved in Mauá et al. [Mauá et al., 2017]:

**Theorem 15** ([Mauá et al., 2017])**.** *Computing lower/upper conditional expectations of a variable in CSPNs takes at most polynomial time when each internal node has at most one parent.*

From Proposition 14 and Theorem 15, we can infer the next corollary.

**Corollary 16.** *Computing lower and upper conditional expectations of a variable in CPSNs learned using LearnCSPN takes at most polynomial time.*

A second result from [Mauá et al., 2017] is also worth noting:

**Algorithm 3** Find best clusters for instance *inst*

---

**Require:** *inst*, set of clusters *nbcs*
**Ensure:** A set of clusters *clusters_for_inst*
 1: Let *bestCLL* be the best current log likelihood of *inst* with a cluster
 2: *bestCLL = newClusterPenalizedLL*
 3: Let *bestCLL_worstcasescenario* be the best current log likelihood of the worst case scenario of *inst* with a cluster
 4: *bestCLL_worstcasescenario = newClusterPenalizedLL*
 5: Let *clusters_for_inst* initialized to an empty set
 6: *Have_found_cluster = false*
 7: **for each** *cluster* **in** *nbcs* **do**
 8:     *cll* ← log likelihood of *inst* in *cluster*
 9:     *worstcase* ← log likelihood of the worst case scenario of *inst* in *cluster*
10:     **if** *cll > bestCLL* **then**
11:         *Have_found_cluster = true*
12:         *bestCLL = cll*
13:         *bestCLL_worstcasescenario = worstcase*
14:         **for each** *clus* **in** *clusters_for_inst* **do**
15:             **if** *clus.LL < bestCLL_worstcasescenario* **then**
16:                 remove *clus* from *clusters_for_inst*
17:             **end if**
18:         **end for**
19:         *clusters_for_inst.add(cluster)*
20:     **else if** *cll > bestCLL_worstcasescenario* **then**
21:         *clusters_for_inst.add(cluster)*
22:     **end if**
23: **end for**
24: **if not** *Have_found_cluster* **then**
25:     *c = newCluster*
26:     *nbcs.add(c)*
27:     *clusters_for_inst.add(c)*
28: **end if**

**Theorem 17** ([Mauá et al., 2017]). *Consider a CSPN $\{S_w : w \in C\}$, where C is the Cartesian product of finitely-generated polytopes $C_i$, one for each sum node i. Computing $\min_w S_w(\lambda)$ and $\max_w S_w(\lambda)$ takes $O(sL)$ time, where s is the number of nodes and arcs in the shared graphical structure and L is an upper bound on the cost of solving a linear program $\min_{w_i} \sum_j c_{ij} w_{ij}$ subject to $w_i \in C_i$.*

Intuitively, this result says that the computational complexity associated with upper log-likelihoods in a tree-shaped CSPN remains linear. In the case of that work, it was due to calling a linear solver on each of the sum nodes, to optimise and propagate values to the root node. And this is all the more true when dealing with extreme points, and so we get that the computational complexity for log-likelihoods is linear in the number of extreme points multiplied by the number of nodes.

**Corollary 18.** *Computing upper/lower log-likelihood of evidence takes $O(n * N)$ time where n is the number of nodes and N is the maximum number of extreme points in the learned CSPN.*

## 6.7 Experimental Evaluations: Additional Details

### 6.7.1 DATASETS

In Table 6, we recall the characteristics of the datasets in terms of number of variables ($\|V\|$), number of instances in the training set (Train), number of instances in the validation set (Valid) and number of instances in the test set (Test).[12]

Table 6: Dataset statistics

| Dataset | $\|V\|$ | Train | Valid | Test |
|---------|------|-------|-------|------|
| NLTCS | 16 | 16181 | 2157 | 3236 |
| Plants | 69 | 17412 | 2321 | 3482 |
| Audio | 100 | 15000 | 2000 | 3000 |
| Jester | 100 | 9000 | 1000 | 4116 |
| Netflix | 100 | 15000 | 2000 | 3000 |
| Retail | 135 | 22041 | 2938 | 4408 |
| Pumsb-star | 163 | 12262 | 1635 | 2452 |
| DNA | 180 | 1600 | 400 | 1186 |
| MSWeb | 294 | 29441 | 3270 | 5000 |
| Book | 500 | 8700 | 1159 | 1739 |
| EachMovie | 500 | 4524 | 1002 | 591 |
| WebKB | 839 | 2803 | 558 | 838 |
| Reuters-52 | 889 | 6532 | 1028 | 1540 |
| 20 Newsgrp. | 910 | 11293 | 3764 | 3764 |
| BBC | 1058 | 1670 | 225 | 330 |
| Ad | 1556 | 2461 | 327 | 491 |

---

12. Out of the 20 datasets present in [Gens and Domingos, 2013], we considered 16 datasets, which span various types of domains, and where variable numbers were as many as 1600, and the number of instances were as many as 28k. We did not consider 4 of them, one of which has about 300k instances, as our evaluations frequently ran out of memory on our machine as we increased the percentage of missing values. The issue is thus solely related to computational power: reasonable subsets of these datasets are not problematic.

### 6.7.2 INFERENCE RESULTS: GRAPH PLOTS

Figure 6 recall results from the paper plus the additional 14 graph plots of the remaining datasets. In the same way, we notice the similar results to the one from the first experiment (the accuracy of the learning algorithm).

## 6.8 Discussion on Related Works

Mauá et al. [Mauá et al., 2017] developed credal sum-product networks, an imprecise extension of sum-product networks, as a motivation to avoid unreliability and overconfidence when performing classification (*e.g.* in handwritten digit recognition tasks). In their work, they consider $\epsilon$-contamination to obtain the credal sets associated to edges of sum nodes in order to evaluate the accuracy of CSPNs in distinguishing between robust and non-robust classifications. More precisely, they look for the value of $\epsilon$ such that the CSPN obtained by local contamination on each sum node results in single classification under maximality. It differs from our work as we learn both the structure and weights of a CSPN based on missing values in the dataset. Intuitively, to compute tasks like classification with a CSPN learned using Algorithm 1, it searches for the optimal CSPN given the evidence and return a set of classes that are maximal. It also differs in the inference process as we focus on extreme points instead of linear programming to find the maximum (or minimum) value to propagate.

As a credal representation [Levi, 1980], open-world probabilistic databases [Ceylan et al., 2016] are also closely related to credal networks. They extend probabilistic databases using the notion of *credal sets*. Their idea is to assume that facts that do not belong to the probabilistic database should not be considered as false, and in this sense, the semantics is open-world. Consequently, these facts are associated with a probability interval $[0, \lambda]$ where $\lambda$ is the threshold probability. This relates to our work in the sense that the probability for missing facts (that would be illustrated by all instances having a missing value for the fact) varies in a closed set $[0, \lambda']$ where the threshold $\lambda'$ is defined by a combination of the weights associated to sum nodes.

SPNs have received a lot of attention recently. For example, in [Hsu et al., 2017] the authors propose an online structure learning technique for SPNs with Gaussian leaves. It begins by assuming that all variables are independent at first. Then the network structure is updated as a stream of data points is processed. They also allow a correlation in the network in the form of a multivariate Gaussian or a mixture distribution. In [Molina et al., 2018, Bueff et al., 2018], the authors tackle the problem of tractable learning and querying in mixed discrete-continuous domains.

There are, of course, other approaches to representing imprecise probabilistic information, such as belief functions [Shafer, 1976, Denoeux, 2011]. Possibilistic frameworks [Zadeh, 1999] have also been introduced to deal with imprecise and more specifically incomplete information. Many works have also been devoted to study the properties and relationships between the two frameworks [Haddad et al., 2017, Benferhat et al., 2017].

Finally, it is worth remarking that open-world semantics and missing values is a major focus in database theory [Libkin, 2016, Console et al., 2017], and it would be interesting to see if our work can be related to those efforts as well.

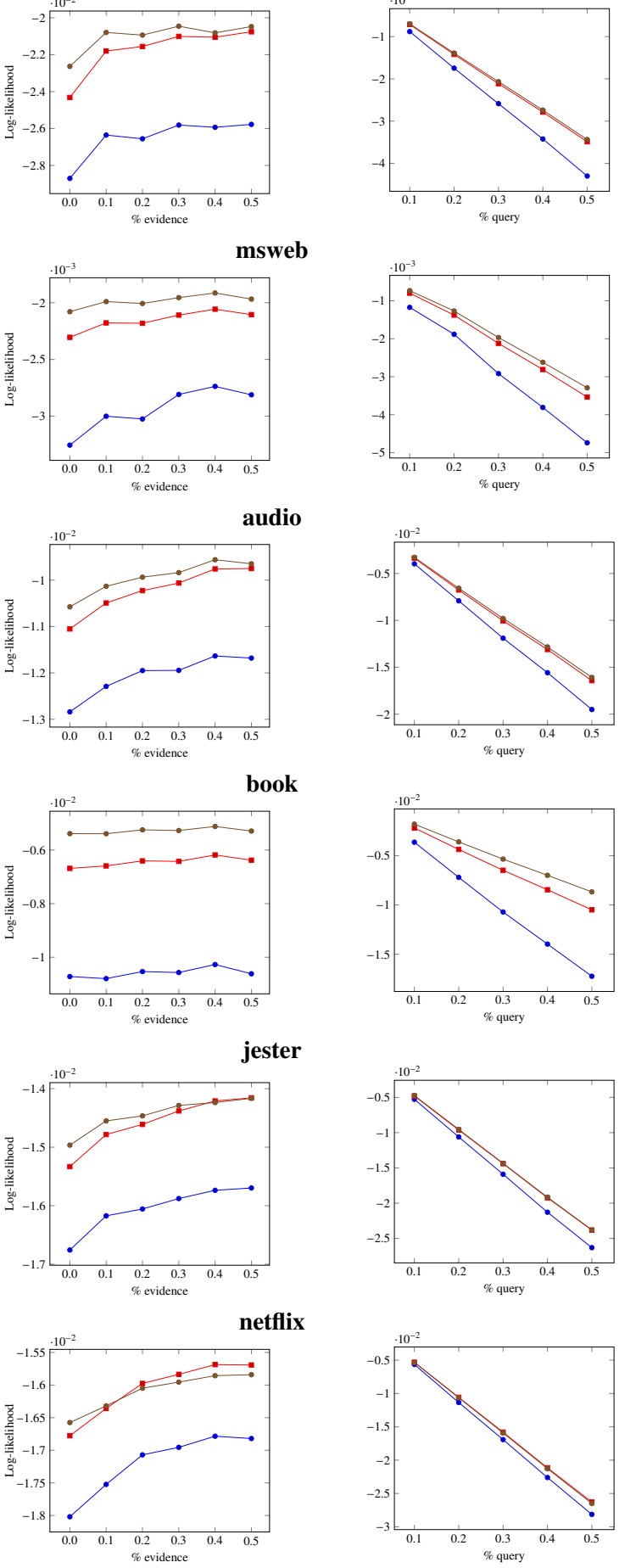

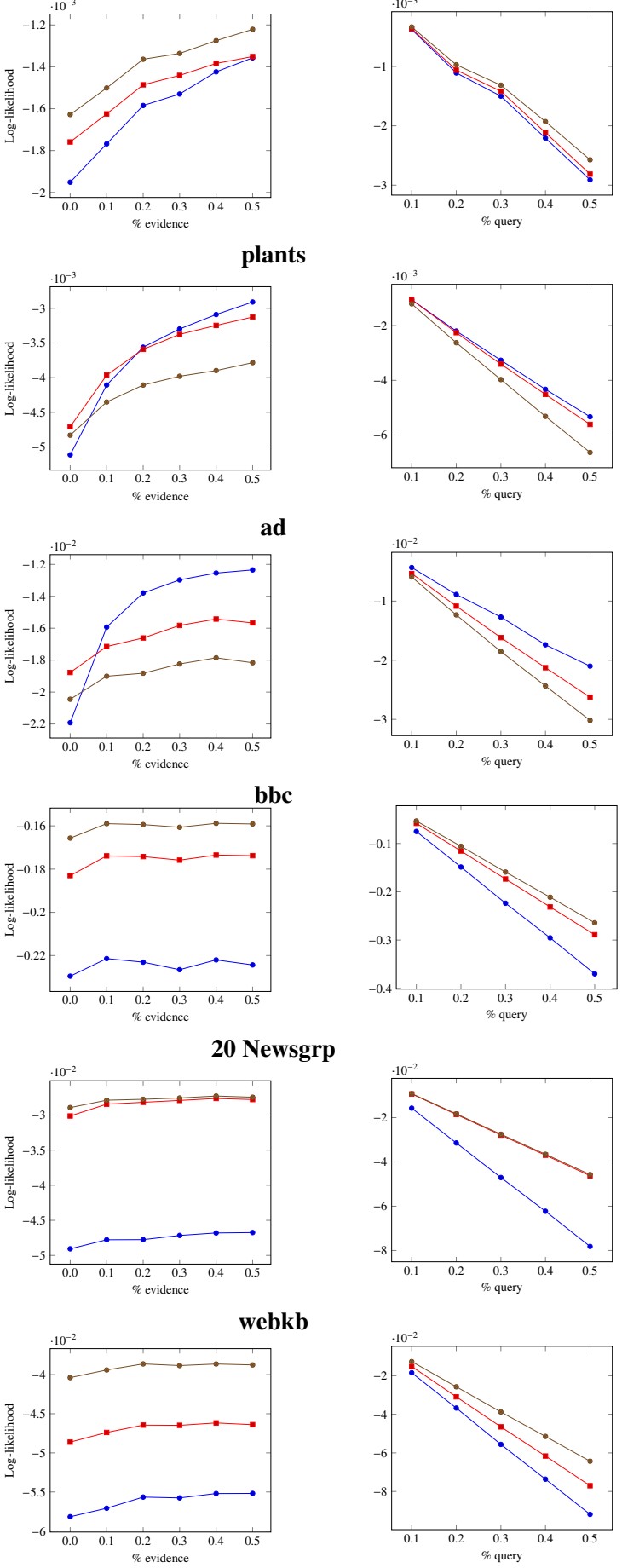

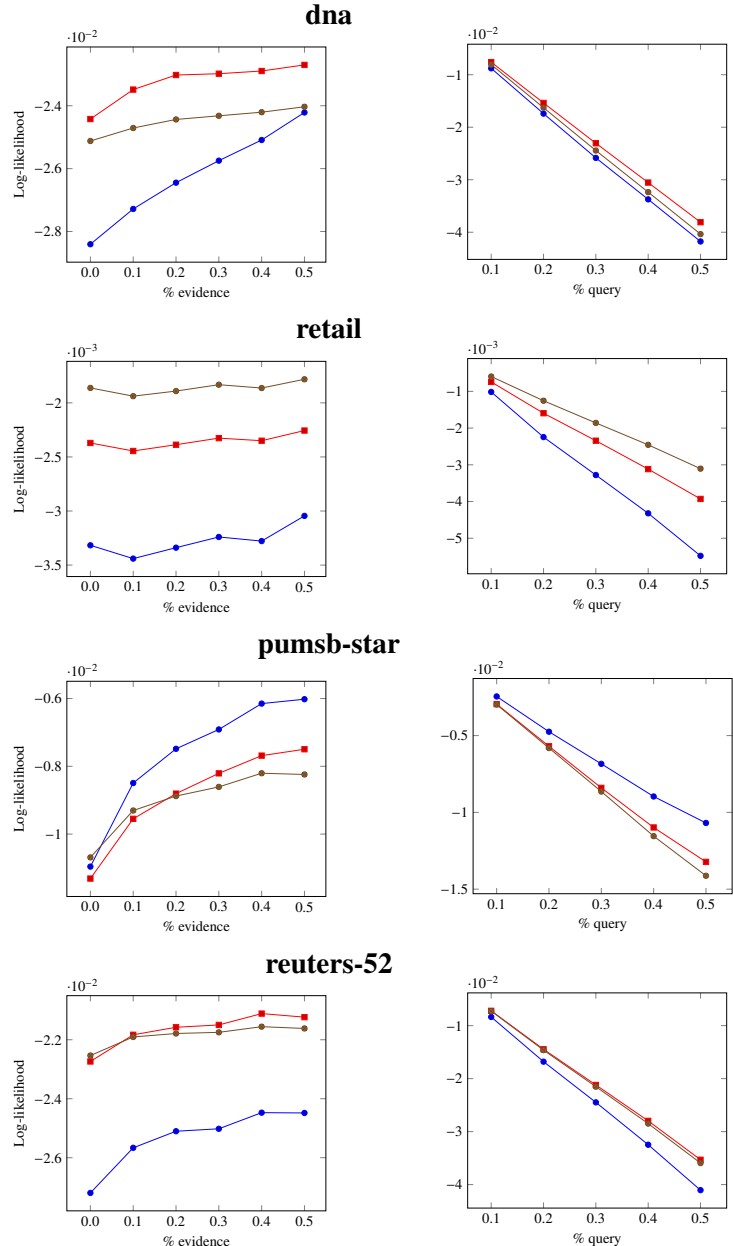

Figure 6: Average log-likelihood normalised by the number of query variables. For each dataset, the left plot fixes the fraction of evidence variables at 30% and varies the fraction of query variables; the right plot fixes query variables at 30% and varies evidence.