# OpenReview forum: "Learning Credal Sum-Product Networks"
_AKBC.ws/2020/Conference — AKBC 2020_

### Official Review · AnonReviewer2 · 2020-03-24
**Interesting direction following rather standard arguments with unclear experimental evaluation**

**Rating:** 5
**Confidence:** 5

**Review:**

The paper revisits Credal SPNs and proposed a learning approach for Credal SPNs in the presence of missing information That is, now the weights on sum nodes vary in closed and convex set and in turn, one gets a imprecise probability model.

Overall, the paper is well written and structured. The main technical contribution are (1) a group-wise independence test and (2) clustering method, both for the credal setting assuming missing data. Specifically, the independence test is a directly application of complete case analysis plus interpreting missing values as contribution to the base population. For the clustering, thee authors should argue why not existing methods for clustering with incomplete data could be use. In any case, the likelihood approach presented also follow the same logic as the independence test. In both cases, the arguments are a little bit hand waving and fluffy. For instance, it is not clear to me what 2is that value that is poorest fit2 (page 6). Still, the clustering is interesting, although as said, a discussion of related work on clustering incomplete data is missing.

The empirical evaluation is interesting and follows the standard protocol for SPN. What i am missing is a repeated argument of why CSPNs are important. Furthermore, the running time should be reported. Also, the authors should provide some insights into the structures learned, also in comparison to the complete data case and the even to the standard SPN setting.  Furthermore, it might be interesting to use Random Credal SPNs based on Random SPNs (Peharz et al. UAI 2019) as a baseline to illustrate the benefit of structure learning. Currently the results just show likelihood. But shouldn't we also consider here the number of parameters? At least getting some numbers here would be appreciated. Also, sincce you consider the CLL, one should also show a discriminatively learned SPN. General, the experimental protocol should be described at sufficient details.  What were the hyperparameters? Was this crossvalidated?

To summarize, nice direction with follows the standard approach for learning SPN  for learning CSPN with using ideas from data imputation. The empirical evaluation is not well presented in the main part. Some missing related work on the clustering with incomplete data.

---

> ### Author Response · Authors · 2020-04-09
> **responses and clarifications**
>
> We thank the reviewer for the feedback. We will attempt to address the concerns raised below, and clarify some of the misunderstandings.
>
> Clustering:
>
> this is an interesting suggestion of leveraging the literature of clustering with missing data, but it is not clear if the application is that immediate for the specific task of constructing interval probabilities. Recall that we can’t appeal to ideas such as imputation and shrinkage, nor is it immediate that assuming properties about marginal distributions for obtaining “better” clusters is necessarily what we want. Indeed, we don’t want to simply construct the best clusters, but rather build clusters that capture the uncertainty present in the data so as to model random variables with imprecise probabilities. Thus, it seems to us that the approach we take with lower and upper bounds is the most sensible and semantically obvious step for building clusters for our setting. For future work, it might be interesting to consider whether other ideas from clustering with incomplete data could be leveraged for faster training.
>
> We can also add a note about related work.
>
> Importance and relevance of CSPNs:
>
> The fundamental characteristic of using imprecise models is to be able to explicitly capture information that we are uncertain about. As we explain in our response to one of the other reviews, CSPNs allows the user to understand that there is incomplete knowledge, and so can use the imprecise probability for assessing the degree of confidence in a returned answer. This has motivated proposals such as [2] among others. Credal semantics can also play a role in bayesian updates: rather than assuming that every entry has zero probability, we might ascribe it a range [0,\lambda], and so if we ever do see observe the entry, we can provide a justified posterior, thereby providing a principled means of handling “new and unseen” observations. This argument is also used in [2].
>
> Other baselines:
>
> We appreciate this point, but the challenge for us (as we responded to the other review) was to make sure the evaluation is semantically justified. Because SPNs appeal to EM to handle missing values, this is one aspect in terms of which we can draw direct comparisons. But even then, there is principle difference — CSPNs allows the user to understand that there is incomplete knowledge as discussed above. In general, we acknowledge the thrust of this point — to have a deeper understanding of the wider applicability of CSPNs, we need a more comprehensive analysis of their use in a wide variety of applications. We hope, nonetheless, that the reviewer acknowledges this is out of scope, as the current paper takes the first step in terms of the foundations and has already run many pages into the appendix.
>
> We are happy to add more details about the experiments, and we will also be releasing the code.

---

### Official Review · AnonReviewer1 · 2020-03-27
**review for learning credal sum product networks**

**Rating:** 6
**Confidence:** 3

**Review:**

In this paper the authors investigate probabilistic representations for learning from incomplete data and specifically investigate credal sum product networks. CSPN are better able to consider data incompleteness which is an important aspect of knowledge bases. The authors perform experiments on a large number of datasets with varying amounts of artificially missing data observing that the optimized log liklihood computed on a learned CSPN generally performed the best.

The paper is generally well written and does a good job of explaining the underlying models and algorithms. The paper is not particularly novel but contains a large number of experiments that could be useful to those interested in probabilistic models in regimes with missing data.

other comments:
- table 4 is a bit busy, there could be a clearer way of presenting and highlighting the relevant results.
- section 4.2 has an occurrence of a  CPSN type-o

---

> ### Author Response · Authors · 2020-04-09
> **response**
>
> Thank you for the feedback. We will attempt to improve the presentation of table 4.
>
> Please see if our comments to the other reviewers' feedback addresses the point about significance/novelty.

---

### Official Review · AnonReviewer3 · 2020-03-28
**A learning algorithm for a type of graphical model, evaluation is a little limited**

**Rating:** 6
**Confidence:** 3

**Review:**

Summary: the paper is presenting a learning algorithm for Credal Sum Product Network (CSPN), a type of graphical model that is tractable (easy to compute partition function), and can encode uncertainty in the network parameters (instead of fixed weights, the network parameters have range of values (or more generally, are defined using a set of convex constraints between them)). Prior work [Maua et al., 2017] introduced CSPNs and provided an inference algorithm, and this paper is the first to propose a learning algorithm for CSPNs.

Pros: first paper to introduce a weight learning algorithm for CSPNs. Evaluation shows better results than Sum Product Network (SPNs)

Cons:
- evaluation is limited in two aspects, baselines and tasks.
1) baselines: the only baseline considered is SPNs, which is a reasonable but old baseline. It would be good to see how well CSPN learning works compared to more recent models, especially that even CSPN's inference evaluation [Maua et al., 2017] was similarly limited.
2) tasks: evaluation avoided large datasts. It excluded the largest of the subtasks (footnote page 21), and evaluating on large scale textual data is left for future work. Even though the motivation for SPN was that its inference is tractable and fast, the proposed learning algorithm for CSPNs seems to be 10x slower than that of SPN and didn't scale to large datasets.

Notes:
- The paper mentioned that CSPN avoids the closed-world assumption, and can work with incomplete examples. I agree with the second but not the first. The proposed learning algorithm takes into account that some instances have unknown values, but it is still assuming that the world only contains the provided list of instances (closed-world assumption).
- The paper use of the term "lifting" seems different from how it is used in Broeck et al., 2011 (doing inference at the first-order level without grounding to predicate logic). This needs to be clarified.

---

> ### Author Response · Authors · 2020-04-09
> **responses and clarifications**
>
> We thank the reviewer for the detailed feedback. We appreciate the concerns raised, and our responses are below.
>
> Baseline:
>
> This is a fair point about the SPN paper being a few years old, but we note that almost all recent work on SPNs do consider LearnSPN as a baseline. For example, see [1] among many others at https://github.com/arranger1044/awesome-spn
>
> [1]: Bayesian Learning of Sum-Product Networks. NeurIPS 2019: 6344-6355
>
>
> The main issue with drawing comparisons is to make sure the evaluation is semantically justified. Because SPNs appeal to EM to handle missing values, this is one aspect in terms of which we can draw direct comparisons. But even then, there is principle difference — CSPNs allows the user to understand that there is incomplete knowledge, and so can use interval probabilities for assessing the degree of confidence in a returned answer. This has motivated proposals such as [2] among others.
>
> [2]: Open-World Probabilistic Databases. KR 2016: 339-348
>
>
> In general, we acknowledge the thrust of this point — to have a deeper understanding of the wider applicability of CSPNs, we need a more comprehensive analysis of their use in a wide variety of applications. We hope, nonetheless, that the reviewer acknowledges this is out of scope, as the current paper takes the first step in terms of the foundations and has already run many pages into the appendix.
>
> Tasks:
>
> We do not find this criticism to be a fair point. We are transparent about the limitations of the model: “One drawback of LearnCSPN is the computational time during learning. This is due to the consideration of an instance in multiple clusters as well as the computation of more values during the clustering process.” Out of the 20 datasets present in [Gens and Domingos, 2013], we considered 16, and the other 4 (indeed very large) ones were dropped after observing that our evaluations frequently ran out of memory on our machine as we increased the percentage of missing values. We didn’t want to augment our computational clusters only for these 4 datasets.
>
> This outcome, in fact, is immediate — any method that needs provably more clusters is bound to be more expensive. The advantage is during inference that has better accuracy. But, as mentioned above, this is not the only reason to use CSPNs. It has added semantic value of knowing that a random variable may have an interval probability, indicating to the user that his confidence in predictions involving said random value should be reduced, on account of missing values. This makes CSPNs an interesting model to consider from a robustness/open-world (in the sense of “open-world probabilistic databases”) viewpoint.  Credal semantics play a role in bayesian updates: rather than assuming that every entry has zero probability, we might ascribe it a range [0,\lambda], and so if we ever do see observe the entry, we can provide a justified posterior, thereby providing a principled means of handling “new and unseen” observations.
>
> Open-world:
>
> This is a fair point, but the community is somewhat divided in terms of its use of these terms. In some circles, closed-world means having one unique (logical) interpretation, which means that every atom is known to be true or false. An open-world model relaxes that. Russell (2015) uses the term open-universe to mean the domain is unbounded. See discussions in [2] above: “Reiter (1978) introduced the open-world assumption (OWA) as the opposite of the CWA. Under the OWA, a set of tuples no longer corresponds to a single interpretation. … OpenPDBs follow a rich literature on interval-based probabilities (Halpern 2003), credal networks (Cozman 2000) and default reason- ing (Reiter 1980).”
>
> We will clarify this point in the paper.
>
> Lifting: we meant in the natural language sense of the word, as in “improve or enhance”. We can remove the use of this term.

---

### Decision · Program_Chairs · 2020-05-01

**Decision:**

Accept

**Comment:**

This paper develops the first structure learning algorithm for Credal SPNs. The paper is somewhat difficult to evaluate, since the credal paradigm is so different from the usual maximum likelihood paradigm, which makes a direct empirical comparison challenging. By providing more detailed information about the uncertainty, the credal approach certainly has some merit, and while upgrading some SPN structure learning heuristics to the credal setting may not be technically challenging, they are done for the first time in this paper. On the other hand, the reviewers did find many ways in which the paper can be improved. Overall, we recommend acceptance. The authors are encouraged to improve the paper as suggested by the reviewers.